# Identification of Key Nucleotide Metabolism Genes in Diabetic Retinopathy Based on Bioinformatics Analysis and Experimental Verification

**DOI:** 10.3390/biology14040409

**Published:** 2025-04-12

**Authors:** Wei Wang, Jianyang Gong

**Affiliations:** 1Department of Ophthalmology, The First Affiliated Hospital of Anhui Medical University, Hefei 230022, China; powerwwang@gmail.com; 2Anhui Public Health Clinical Center, Hefei 230022, China

**Keywords:** diabetic retinopathy, nucleotide metabolism, biomarker, transcription factor, RT-PCR

## Abstract

The mechanisms involved in nucleotide metabolism in DR were investigated based on RNA sequencing data and single-cell data in diabetic retinopathy (DR). The results demonstrate that HMOX1, TLR4, and ACE are core genes in DR. These three biomarkers were downregulated in DR. According to the GSVA results, interferon alpha response, IL6_JAK_STAT3 signaling, and apoptosis were activated in the DR group. Based on transcription factor prediction, TLR4 and HMOX1 may be target genes of USF2. Importantly, HMOX1–Stannsoporfin (score = −10.1 kcal/mol) and cilazapril–ACE (score = −9.0 kcal/mol) had high affinities.

## 1. Introduction

Diabetic retinopathy (DR)—a common eye disease that can lead to blindness—is one of the most common microvascular complications in people with diabetes mellitus [1,2]. Most patients will begin to exhibit signs of diabetic retinopathy within 20 years after developing diabetes [3], and approximately half of the patients with untreated proliferative retinopathy will go blind within 5 years [4]. Traditionally, a diagnosis of diabetic retinopathy was established via fundus examination, which is highly subjective and often overlooks early-stage lesions. Biomarkers can reflect the disease state at the molecular level, providing objective indicators to facilitate early and accurate diagnosis of diabetic retinopathy. Bioinformatics analysis can integrate massive biological data, excavate key molecules and pathways related to diabetic retinopathy, and facilitate the discovery of new diagnostic markers [5,6,7].

Despite recent advancements in the prevention and treatment of diabetic retinopathy, numerous challenges remain. Accurately predicting and controlling the onset and progression of diabetic retinopathy is highly challenging, and when it comes to treatment, the complex pathological changes which occur at different stages of the disease make it difficult to ensure therapeutic efficacy, and the recurrence rate remains high [8,9,10]. Therefore, there is an urgent need to thoroughly investigate the molecular mechanisms of DR and identify new diagnostic biomarkers and therapeutic targets to enhance the effectiveness of prevention and treatment, thereby alleviating the burden on society and families.

Nucleotide metabolism plays a crucial role in cellular functions. Firstly, ATP (adenosine triphosphate), as the primary energy carrier of the cell, is generated through energy metabolic pathways such as glycolysis, the citric acid cycle, and oxidative phosphorylation, fueling various cellular activities. Simultaneously, nucleotide metabolism is essential for maintaining the balance of ATP and other nucleotides, ensuring a stable supply of cellular energy. Secondly, during signal transduction processes, nucleotide derivatives such as cyclic adenosine monophosphate (cAMP) and cyclic guanosine monophosphate (cGMP) act as secondary messengers, participating in intracellular signal transmission [11,12,13,14]. Furthermore, nucleotide metabolism is involved in the synthesis and degradation of DNA and RNA, ensuring cell proliferation and the transmission of genetic information. Nucleotide metabolism occupies a vital position within the immune–metabolic network, influencing the function of immune cells and the regulation of inflammatory responses by modulating cellular energy supply and biosynthesis [15].

In diabetic patients, hyperglycemia-induced oxidative stress is key to the pathogenesis of DR [16]. Oxidative stress not only directly damages the structure and function of retinal cells, leading to cell apoptosis and retinal vascular pathology, but it also activates inflammatory responses, promoting the progression of diabetic retinopathy [16,17]. Inflammation, another important pathological mechanism of diabetic retinopathy, exacerbates retinal damage and vascular abnormalities through the release of pro-inflammatory cytokines [18]. Several nucleotide metabolism-related genes may be involved in regulating inflammatory pathways, influencing the progression of diabetic retinopathy [15]. Therefore, nucleotide metabolism-related genes may play significant roles in DR, but the specific mechanisms involved are in need of further investigation.

This study aimed to identify and characterize key nucleotide metabolism-related genes involved in diabetic retinopathy (DR) through bioinformatics analysis and experimental validation. To achieve this, we integrated multiple datasets from GEO and conducted comprehensive bioinformatics analyses to investigate the potential roles of nucleotide metabolism-related genes in DR. After gene screening, three genes—HMOX1, TLR4, and ACE—were selected as core genes based on their diagnostic performance (Area Under the Curve > 0.8). In addition, we performed gene set variation analysis (GSVA) to explore the potential biological function, and potential drug prediction analysis to identify possible therapeutic agents targeting these genes. Immune cell infiltration analysis was also conducted to explore the involvement of immune-related pathways in the pathological process of DR. These analyses collectively highlight the relevance of these key genes in DR and their potential as novel therapeutic targets. Our findings provide a foundation for future research aimed at developing personalized medical approaches for the effective management and treatment of diabetic retinopathy.

## 2. Materials and Methods

### 2.1. Data Acquisition and Processing

We leveraged the publicly accessible NCBI Gene Expression Omnibus (GEO) database (http://www.ncbi.nlm.nih.gov/geo/, (accessed on 20 November 2024)) to retrieve DR-related RNA sequencing (RNA-seq) datasets. The single-cell sequencing dataset GSE165784, containing six DR samples, was obtained. GSE102485 (based on the GPL18573 platform) consisted of neovascular proliferative membrane specimens from 21 DR patients and 3 controls, and was employed to define differentially expressed genes (DEGs). Another dataset, GSE60436 (also accessed via the GPL18573 platform), includes the fibrovascular membranes of 6 patients with DR and 3 controls, and was utilized to validate gene expression levels. Following data merging, we normalized and corrected potential batch effects using the R package sva (v. 3.35.2). Additionally, 882 nucleotide metabolism-related genes (NM-RGs) were selected from the GeneCards database (version 5.23) (https://www.genecards.org/, (accessed on 20 November 2024)) and then annotated based on nucleotide metabolic pathways [19] (Appendix A).

### 2.2. Identification of DEGs

Within the GSE102485 dataset, we utilized the R package DESeq2 (v. 1.34.0) to identify DEGs between diabetic retinopathy and normal samples (|log2FC| > 0.5, *p* value < 0.05). Volcano and heatmap visualizations of the DEGs were created using the R package ggplot2 (v. 3.3.6) [20] and heatmap (v. 1.1.9) [21], respectively. For the volcano plot, the top ten upregulated and top ten downregulated genes were highlighted, while the heatmap depicted the expression patterns of the same subsets of DEGs. Subsequently, the differentially expressed NM-RGs (DE-NMRGs) were obtained by identifying the intersection between the DEGs and NM-RGs. The results were displayed in the form of a Venn diagram, using the R package ggVenn (v. 0.1.10) package [22].

### 2.3. GO and KEGG Enrichment Analysis

To learn more of the functions of DE-NMRGs, we utilized the R package clusterProfiler (v. 4.7.1.003) [23] for GO and KEGG pathway enrichment analyses (*p* value < 0.05). The results were sorted in descending order based on the number of genes involved, and the top three GO terms for each domain were presented, retrospectively. The top five KEGG pathways based on the number of enriched genes were selected and visualized.

### 2.4. Construction of Protein–Protein Interaction (PPI) Network and Selection of Core Genes

The interaction between DE-NMRGs at the protein level was explored via the STRING online platform (http://string-db.org, (accessed on 30 December 2024)), resulting in the development of a PPI network with a combined score > 0.4. The isolated nodes (i.e., those with no interacting genes) were excluded in this step, and the network was visualized using Cytoscape (v. 3.9.1) software [24]. Subsequently, the CytoHubba plugin was employed to evaluate the importance of each DE-NMRG using five algorithms: MCC, EPC, MNC, Degree and DMNC. The top ten genes from each algorithm were selected using the R package VennDiagram (v. 1.7.3) (https://CraN.R-project.org/package=VennDiagram, (accessed on 30 December 2024)) to identify core genes among the most prevalent examples from each algorithm.

### 2.5. Identification and Validation of Biomarkers

The diagnostic capability of core genes was evaluated in GSE102485 via the R package pROC (v. 1.18.0) [25]. Meanwhile, Receiver Operating Characteristic (ROC) curves were created, and the candidate biomarkers were selected (Area Under the Curve (AUC) > 0.8). To validate the expression levels of the candidate biomarkers, an independent validation set (GSE60436) was introduced. The biomarkers were required to meet the significant expression differences between DR and controls in both the GSE102485 and GSE60436 sets (Wilcoxon rank-sum test, *p* value < 0.05).

### 2.6. Chromosome and Subcellular Localization

To investigate the chromosomal locations of the biomarkers, we downloaded the positional information of the relevant genes from the ENSEMBL (https://www.ensembl.org, (accessed on 3 January 2025)) database. The R package RCircos (v. 1.2.2) [26] was utilized to generate chromosomal position maps, illustrating the specific genomic locations of the biomarkers. To elucidate the functions and mechanisms of action of the biomarkers, we obtained FASTA sequences of said biomarkers from the NCBI website (https://www.ncbi.nlm.nih.gov/, (accessed on 10 January 2025)). These sequences were input into the mRNALocater database [27] to predict subcellular localization.

### 2.7. Correlation Analysis and Gene–Gene Interaction (GGI) Network

To evaluate the correlations between different biomarkers, we utilized the R package psych to perform Spearman correlation analysis of GSE102485. Afterwards, the GeneMANIA database (http://www.genemania.org/, (accessed on 21 January 2025)) was employed to build a GGI network based on the biomarkers and the top 20 genes with the strongest functional similarity within their biomarkers.

### 2.8. Gene Set Enrichment Analysis (GSEA)

We employed the single-gene GSEA approach to identify the regulatory pathways and biological functions associated with the expression of target biomarkers. We began by calculating the Spearman correlation coefficients between the biomarkers and all other genes in GSE102485 and generating a list of correlation coefficients in descending order. Subsequently, we downloaded c2.cp.kegg.v2023.1.Hs.symbols.gmt from the Molecular Signatures Database (MSigDB) for analysis; the R package clusterProfiler [28] (v. 4.7.1.003) was used for the subsequent evaluations (*p* value < 0.05). To further investigate the relationships between the biomarkers and enriched pathway, we identified the union set of the top five pathways for each biomarker. We then conducted Spearman correlation analysis of GSE102485 to assess the correlations between the genes within the top five pathways enriched by the biomarkers (|cor| > 0.3, *p* value < 0.05).

### 2.9. Gene Set Variation Analysis (GSVA)

We utilized GSVA to investigate the alterations in pathway activities between the two groups within the GSE102485 dataset. Firstly, we downloaded the background gene set h.all.v2023.1.Hs.symbols.gmt from the MSigDB database. The GSVA score of each sample and differential analysis of the GSVA scores between the DR and controls in GSE102485 (t ≠ 1, *p* value < 0.05) were obtained using the GSVA (v. 1.46.0) package [29].

### 2.10. Immuno-Infiltration Analysis

To investigate the differences in immune cell infiltration between the DR and control groups within GSE102485, we utilized the CIBERSORT algorithm [30] to calculate the infiltration proportions of 22 common immune cells. The immune cells with a relative percentage of 0 or less than 50% of the cells in the sample were excluded, and the infiltration differences were compared between the DR and controls (*p* value < 0.05). Finally, Spearman’s correlation analysis was employed to explore the associations between the biomarkers and immune cells.

### 2.11. miRNA and Transcription Factor (TF) Prediction

We performed miRNA prediction for biomarkers using the R package multiMiR (v. 1.20.0) [31] package in conjunction with the ElMMO database (http://mirtoolsgallery.tech/mirtoolsgallery/node/1098, (accessed on 11 February 2025)). Based on the prediction results, miRNAs associated with each of the biomarkers were selected. To explore TFs that target and regulate the biomarkers, we employed the ChEA3 database (https://maayanlab.cloud/chea3, (accessed on 11 February 2025)) for TF prediction. TFs associated with the regulation of biomarkers were selected; only those with a score of >600 were eligible.

### 2.12. Drug Prediction and Molecular Docking

To explore potential treatments for DR, we utilized the Drug–Gene Interaction Database (DGIdb, www.dgidb.org, (accessed on 15 January 2025)) for drug prediction. Subsequently, the molecular structures of the drugs were obtained from the PubChem database (https://pubchem.ncbi.nlm.nih.gov/, (accessed on 15 January 2025)). The protein structures of biomarkers were retrieved from the UniProt database (https://www.uniprot.org/, (accessed on 15 January 2025)). Then, molecular docking between the biomarkers and drugs was performed using the CB-Dock database (https://cadd.labshare.cn/cb-dock/php/blinddock.php, (accessed on 15 January 2025)). Docking scores (Interaction Score) below −5 kcal/mol were indicative of high binding affinity between the drug and the target protein. The binding capabilities of the drugs with the biomarkers were assessed by comparing their docking scores. The results of the molecular docking experiments were visualized using PyMOL (v. 3.0) software [32].

### 2.13. scRNA-Seq Analysis

Single-cell RNA sequencing data (GSE165784), comprising six DR samples, were downloaded from the GEO database and analyzed using the R package Seurat. Quality-control steps included the removal of cells with >10% mitochondrial gene content, fewer than three detected genes, or more than 8000 expressed genes. Gene expression data were normalized using the *NormalizeData* function, and principal component analysis (PCA) was performed based on the top 2000 highly variable genes to extract the top 20 principal components. For unsupervised clustering of cell subpopulations, the *FindNeighbors*, *FindClusters* (resolution = 0.4), and *RunTSNE* functions were applied. Marker genes for each cluster were identified using the *FindAllMarkers* function with an adjusted *p*-value < 0.01 and |log2 fold change| > 1. Cell types were annotated based on previously published work [33]. Cell–cell communication was further analyzed using the R package CellChat.

### 2.14. RT-PCR

In this study, whole-blood specimens from 5 DR patients and 5 healthy controls were collected from the North District of the First Affiliated Hospital of Anhui Medical University. Subsequently, the TRIzol Reagent (Solarbio, Beijing, China) was applied for the extraction of total RNA. Next, the total RNA was reverse-transcribed into cDNA using the SuperScript IV VILO SuperMix (Solarbio, Beijing, China). The PowerTrack SYBR Green Master Mix (Solarbio, Beijing, China) was used for the qPCR system (the primer sequences are presented in Table 1). The qPCR was performed on the BIO-RAD CFX96TM PCR System. Three technical replicates were utilized in the study, and GAPDH was used as the endogenous control. Finally, the relative expression levels were estimated and compared via 2-ΔΔCT and a t-test (*p* value < 0.05).

### 2.15. Statistical Analysis

All statistical analyses were performed using R (v. 4.2.2). The R package ggplot (v. 3.3.6) was used for violin plot and bar plot visualization. All networks were visualized with Cytoscape (v. 3.9.1).

## 3. Results

### 3.1. Screening and Functional Enrichment Analysis of DE-NMRGs

A total of 1359 DEGs were identified in GSE102485, of which 1119 were upregulated and 240 were downregulated (Figure 1A,B). We intersected the 1359 DEGs with the 882 NM-RGs to obtain 48 DE-NMRGs (Figure 1C). GO analysis indicated that the DE-NMRG genes became enriched in response to oxidative stress, xenobiotic stimulus, lipopolysaccharides, etc. (Figure 1D; Appendix A). Likewise, KEGG analysis revealed that DE-NMRGs were enriched in the AGE-RAGE signaling pathway, the IL-17 signaling pathway, the TNF signaling pathway, etc., in patients with diabetes-related complications (Figure 1E; Appendix A). Moreover, a PPI network was constructed based on these DE-NMRGs, including 40 nodes and 156 edges (Figure 1F).

### 3.2. Identification and Assessment of All Biomarkers for DR

Using the CytoHubba plug-in, the top ten genes of each algorithm were selected from the 48 DE-NMRGs (Appendix A). Three core genes (HMOX1, TLR4, and ACE) were determined via overlapping (Figure 2A). The AUC values of these core genes were greater than 0.9 (AUC of HMOX1, 0.968; AUC of TLR4, 0.968; and AUC of ACE, 0.937), demonstrating their superior diagnostic capabilities (Figure 2B). Additionally, we compared these three genes with published biomarkers [34,35,36], which showed lower AUC values compared to HMOX1, TLR4, and ACE (Appendix A). In GSE102485 and GSE60436, all three genes were downregulated significantly in the DR group, and they were thus selected as biomarkers for DR (Figure 2C,D). According to the location analysis, HMOX1, TLR4, and ACE were found on chromosomes 22, 9, and 17, respectively (Figure 2E). These three biomarkers were mainly highly expressed in the cytoplasm (Figure 2F and Appendix A).

### 3.3. Correlation Analysis and Function Exploration of Biomarkers

In GSE102485, TLR4 was positively correlated with ACE and HMOX1 (cor = 0.37 for ACE and 0.21 for HMOX1, Figure 3A,B). There were no significant correlations between HMOX1 and ACE (cor = −0.09, Appendix A). Afterwards, a gene–gene interaction network was built based on biomarkers and the top 20 related genes, which mainly interacted physically (Figure 3C). Based on the GSEA analysis, HMOX1, TLR4, and ACE were enriched into 74, 105, and 105 KEGG pathways, respectively (Appendix A), and co-enriched into leishmania aethiopica, lysosome, ribosome, etc. (Figure 3D and Appendix A). Moreover, there were eight KEGG pathways in the union set, and the gene pathways analysis revealed that the biomarkers had positive correlations with these pathways (Figure 3E). In the GSVA, the interferon alpha response, IL6_JAK_STAT3 signaling, and apoptosis were activated in the DR group (Figure 3F).

### 3.4. Correlation of Biomarkers and Immune Cells

To investigate the immune microenvironment in diabetic retinopathy (DR) patients, we assessed the relative proportions of 22 immune cell types using the CIBERSORT algorithm (Figure 4A). Following standard filtering, 13 immune cell types were retained for further differential analysis. Notably, activated mast cells were significantly reduced, while neutrophil levels were significantly elevated in the DR group (Figure 4B). Furthermore, activated mast cells showed significant negative correlations with all three biomarkers (Figure 4C).

### 3.5. Molecular Regulatory and Drug–Gene Networks

Using the EIMMo database, we identified eight predicted miRNAs, forming three isolated networks. Additionally, the ChEA3 database predicted 123 transcription factors (TFs) associated with the three biomarkers (score > 600). Notably, USF2 may regulate both TLR4 and HMOX1, while GATA4 may regulate both ACE and HMOX1. Based on these findings, we constructed a TF-mRNA-miRNA regulatory network (Figure 5A).

Using the DGIdb database, we identified potential drug candidates for three biomarkers: HMOX1 (9 drugs, with Stannsoporfin achieving the highest score), TLR4 (22 drugs, with the pertussis vaccine scoring the highest), and ACE (59 drugs, with cilazapril scoring the highest). Figure 5B presents the five drugs with the top interaction score for each biomarker. To evaluate the binding affinity between the biomarkers and their predicted drugs, we selected the top-scoring drug of each biomarker for molecular docking. However, as the molecular structure of the pertussis vaccine was not available in the PubChem database, only HMOX1–Stannsoporfin and ACE–cilazapril underwent further analysis, achieving docking scores of −10.1 kcal/mol and −9.0 kcal/mol, respectively (Appendix A), indicating strong binding affinities (Figure 5C,D).

### 3.6. Single-Cell Analysis Re-Indicated That Nucleotide Metabolism Interacts with the Immune System in Diabetic Retinopathy

The gene expression profiles of 11,436 cells from six DR samples were obtained from the GSE165784 dataset. As shown in Figure 6A and Appendix A, the sequencing depth, number of detected genes, and normalization of selected data were rationalized. The top 2000 highly variable genes were selected for further analysis (Figure 6B). We used RunPCA function to reduce the dimensionality, and 14 clusters were identified (Figure 6C,D and Appendix A). The top five marker genes in each cluster are presented in the heatmap (Figure 6D). Subsequently, we manually annotated each cell cluster, and six cell types (astrocytes, B cells, CD8+ T cells, endothelial cells, macrophages, and monocytes) were identified and visualized (Figure 6E). The proportion of each cell type is depicted in Figure 6F.

Figure 7A shows the distribution and expression of biomarkers in different cell types. The biomarkers were highly expressed in the macrophages and monocytes compared with other immune cell types. Additionally, cell–cell interaction was utilized to investigate the communication between different cell types. Astrocytes exhibited strong interactions with endothelial cells, macrophages, and monocytes, suggesting that they play crucial roles in the immune response and inflammatory processes in DR (Figure 7B,C and Appendix A).

### 3.7. Validation of Biomarker Expression via RT-PCR

To further validate the expression of hub genes in DR, we performed an RT-PCR assay. As shown in Figure 8, both HMOX1 and ACE were significantly underexpressed in the DR group, while TLR4 was significantly overexpressed in the DR group (*p* value < 0.05). The expression trends of ACE and HMOX1 in the experiment were consistent with those in the previous transcriptome data.

## 4. Discussion

Diabetic retinopathy is one of the most common and severe microvascular complications of diabetes, potentially leading to impaired vision or even blindness. Nucleotide metabolism plays a crucial role in cellular functions, encompassing key physiological processes, such as energy supply, signal transduction, and DNA synthesis [37]. Existing studies suggest that disruptions in nucleotide metabolism may be associated with the pathogenesis of diabetic retinopathy by affecting the normal metabolism and functions of retinal cells, thereby participating in the pathological development of diabetic retinopathy. This association provides new insights and directions for research into and treatment of DR [38,39].

In this study, we conducted a bioinformatics analysis of the GSE102485 dataset, screening out 48 DE-NMRGs. Further analysis involving PPI network construction and multiple algorithms identified HMOX1, TLR4, and ACE as biomarkers of diabetic retinopathy. Previous studies have also recognized HMOX1 and TLR4 as diabetic retinopathy biomarkers [40,41]. Compared to other known biomarkers, these three genes exhibited superior AUC values, demonstrating high predictive accuracy for diabetic retinopathy (AUCs > 0.8) [17,36,37,38]. Furthermore, these genes were significantly associated with immune responses, oxidative stress, and metabolic pathways, and demonstrated high predictive accuracy for the disease (AUCs > 0.8). Additionally, drug prediction analysis indicated that Stannsoporfin and cilazapril might serve as potential therapeutic agents, targeting these key genes.

Under normal physiological conditions, HMOX1 catalyzes the degradation of heme to produce iron ions, carbon monoxide, and biliverdin (which can be subsequently converted to bilirubin), which is crucial for maintaining intracellular homeostasis and various metabolic processes, including nucleotide metabolism. Iron ions, as cofactors for many key enzymes involved in nucleotide synthesis and metabolism, play an indispensable role in this metabolic process [42]. For example, ribonucleotide reductase requires iron ions to maintain its activity, enabling the conversion of ribonucleotides to deoxyribonucleotides, a key step in DNA synthesis. However, during the development of diabetic retinopathy, various factors, such as oxidative stress and inflammatory responses induced by long-term hyperglycemia, can lead to downregulation of HMOX1 expression. The decrease in HMOX1 expression reduces the production of iron ions, altering the activity of key enzymes, such as ribonucleotide reductase, and interfering with the normal synthesis and metabolism of nucleotides [43]. This not only affects the proliferation and repair ability of retinal cells, as cell division requires sufficient nucleotides for DNA synthesis, but also may disrupt the stability of intracellular nucleic acids, making retinal cells more vulnerable to damage. In addition, the downregulation of HMOX1 may also weaken its antioxidant function. Under normal circumstances, the bilirubin produced by HMOX1 has antioxidant properties, which can neutralize the reactive oxygen species (ROS) generated in cells, maintain the intracellular redox balance, and provide a stable environment for nucleotide metabolism. When HMOX1 is downregulated, the production of bilirubin decreases, and the accumulation of ROS in cells increases. ROS can attack nucleotides, causing structural damage, further affecting nucleotide metabolism, and ultimately exacerbating the development of diabetic retinopathy [44].

Diabetic retinopathy causes multiple abnormalities, such as hyperglycemia, oxidative stress, and inflammatory responses. These factors promote an increase in the release of damage-associated molecular patterns (DAMPs), such as high-mobility group box 1 protein (HMGB1), etc. As ligands of TLR4, they can activate the TLR4 signaling pathway [45]. Once activated, TLR4 has a significant impact on nucleotide metabolism through a series of signal transduction mechanisms. The activated TLR4 signaling pathway can upregulate the expression of some of the key enzymes involved in nucleotide synthesis. For example, by activating the nuclear factor-κB (NF-κB) signaling pathway, it promotes changes in gene transcription regulation, thereby increasing the expression of phosphoribosyl pyrophosphate synthetase (PRPS). PRPS is a key enzyme in the initial stage of purine nucleotide synthesis, and the upregulation of its expression will accelerate the synthesis of purine nucleotides [46]. This may represent an adaptive response from the body, which allows it to meet the nucleic acid requirements for immune cell proliferation and cytokine synthesis in response to DR-induced cell damage and inflammatory responses. In terms of pyrimidine nucleotide metabolism, the upregulation of TLR4 also plays a role. It may regulate the synthesis rate of pyrimidine nucleotides by affecting the activity of key enzymes in the pyrimidine synthesis pathway, such as aspartate transcarbamoylase (ATCase) [47]. Meanwhile, in the nucleotide salvage synthesis pathway, the activation of the TLR4 signaling pathway may enhance the activity of related enzymes, enabling cells to more effectively utilize free bases and nucleosides to synthesize nucleotides to meet the additional nucleotide demands of stressed cells [48]. However, the continuous and excessive upregulation of TLR4 also has negative effects. Overactivation of the TLR4 signaling pathway may lead to an imbalance in the nucleotide metabolism. The large-scale synthesis of nucleotides will consume excessive amounts of cellular energy and metabolic substrates, inducing intracellular energy metabolism disorders. Moreover, abnormal nucleotide metabolism may further exacerbate the inflammatory response and oxidative stress, creating a vicious cycle and aggravating the pathological damage inflicted by diabetic retinopathy [49,50].

ACE plays a key role in maintaining the body’s homeostasis through the classic renin–angiotensin system (RAS) [51]. Although ACE is not directly involved in the synthesis and decomposition reactions of nucleotides, it has important regulatory effects on cell growth, proliferation, and differentiation, which are closely related to nucleotide metabolism. When diabetic retinopathy occurs, various factors lead to the downregulation of ACE expression. On the one hand, hyperglycemia-induced oxidative stress and inflammatory responses can interfere with the intracellular signal transduction pathway, inhibiting the transcription and translation of the ACE gene and reducing its expression level. On the other hand, long-term exposure to a high-glucose environment may affect the upstream molecular mechanisms that regulate ACE expression in the retinal tissue, further promoting the downregulation of ACE. This downregulation has a significant impact on nucleotide metabolism [52]. Due to the decrease in the ACE, the conversion of angiotensin I to angiotensin II (Ang II) is restricted, resulting in a decrease in the production of Ang II. Ang II plays an important role in the regulation of cell proliferation, and a reduction in its prevalence will weaken the signal. The cell proliferation process is highly dependent on nucleotide synthesis, as DNA replication requires a large number of nucleotides as raw materials. Therefore, the downregulation of ACE indirectly leads to a decrease in the cell’s demand for nucleotides, reducing the activity of key enzymes involved in nucleotide synthesis, such as phosphoribosyl pyrophosphate synthetase (PRPS) and ribonucleotide reductase (RNR), and ultimately affecting the nucleotide synthesis rate [53]. In addition, the downregulation of ACE may also indirectly affect nucleotide metabolism by influencing intracellular energy metabolism. Ang II regulates the contraction and relaxation of blood vessels, thereby affecting tissue blood perfusion and oxygen supply. When the downregulation of ACE leads to insufficient production of Ang II, the regulatory function of retinal blood vessels is impaired, which may cause local ischemia and hypoxia in the retina. In this case, the cell’s energy metabolism shifts from aerobic respiration to anaerobic glycolysis, reducing the energy production efficiency. Since nucleotide synthesis requires sufficient energy supply, the disorder of the energy metabolism will further inhibit nucleotide synthesis, affecting the normal function and proliferation of retinal cells and exacerbating the development of diabetic retinopathy [54].

Stannsoporfin has a high binding affinity for HMOX1 (binding energy: −10.1 kcal/mol) and can form a stable binding state. After binding to HMOX1, Stannsoporfin can inhibit its activity; prevent heme from entering the active site; reduce harmful metabolites; alleviate oxidative stress; relieve cell damage; protect mitochondrial function; maintain energy metabolism; and indirectly block the vicious cycle of “oxidative stress–cytokine release–immune cell infiltration”, reduce NF-κB activation, decrease the expression of cytokines, inhibit the recruitment and activation of immune cells, relieve retinal inflammation and edema, and delay the progression of diabetic retinopathy [55].

The Pertussis Vaccine inhibits NF-κB activation; reduces the release of inflammatory factors such as TNF-α and IL-6; and regulates the cytokine balance and promotes the secretion of IL-10 for anti-inflammation. It also inhibits the overexpression of VEGF, reduces neovascularization, maintains the integrity of vascular endothelial cells, and reduces vascular leakage. It activates the Nrf2 pathway, upregulates the expression of antioxidant enzymes such as HMOX1 and SOD, and scavenges ROS. It also promotes the polarization of macrophages into M2 type for anti-inflammation and the proliferation of Tregs to inhibit the immune response, and it inhibits the differentiation of Th17 cells to alleviate inflammation [56].

Cilazapril inhibits ACE, reduces the production of angiotensin II, dilates retinal blood vessels, and improves blood perfusion; it also increases the amount of angiotensin-(1–7), antagonizes angiotensin II, and improves vascular function. Inhibiting ACE reduces angiotensin II, thereby inhibiting the NF-κB pathway and reducing the number of cytokines; regulates immune cell infiltration; and relieves the inflammatory microenvironment. Additionally, it inhibits the abnormal proliferation of vascular endothelial cells mediated by angiotensin II, prevents neovascularization, reduces angiotensin II production, activates the survival signaling pathway, and inhibits retinal cell apoptosis. It inhibits angiotensin II production, reduces NADPH oxidase activity, and decreases the generation of ROS; it also upregulates the expression of antioxidant enzymes such as SOD and GPX and enhances the antioxidant capacity.

Although we have identified the critical roles of HMOX1, TLR4, and ACE in diabetic retinopathy through bioinformatics analysis and experimental validation, this study still has several limitations. Some of our conclusions lack sufficient experimental verification, particularly in terms of functional confirmation in in vivo models. Future research should validate these findings in larger cohorts and conduct additional in vitro and in vivo experiments, including gene knockdown or overexpression in retinal endothelial cells and diabetic mouse models, to comprehensively elucidate the specific roles of HMOX1, TLR4, and ACE in the pathogenesis of diabetic retinopathy.

Through comprehensive bioinformatics analysis methods, we identified key nucleotide metabolism-related genes, such as HMOX1, TLR4, and ACE, and thoroughly explored their relationship with the pathogenesis of DR. This laid a solid foundation for subsequent research and deepened our understanding of the complex pathological mechanisms of DR. These findings will not only assist with the early diagnosis of DR, with key genes serving as potential biomarkers for disease state monitoring, but will also inform the development of targeted intervention strategies for prevention and treatment. The potential therapeutic drugs identified in this study represent possible treatment options, which are expected to improve the prognosis of diabetic retinopathy patients, enhance their quality of life, and advance the overall diagnostic and therapeutic levels of diabetic retinopathy.

## 5. Conclusions

In conclusion, ACE, HMOX1, and ACE have been identified as promising potential diagnostic biomarkers associated with nucleotide metabolism in diabetic retinopathy. These findings not only deepen our understanding of the molecular mechanisms underlying diabetic retinopathy but also hold significant translational potential. The identified biomarkers may contribute to the development of early diagnostic tools and targeted therapies, aiding in more precise and personalized management of DR. Future research should focus on validating these targets through in vitro and in vivo experiments, exploring their functional roles in retinal pathology, and assessing their potential in clinical applications and drug development.

## Figures and Tables

**Figure 1 biology-14-00409-f001:**
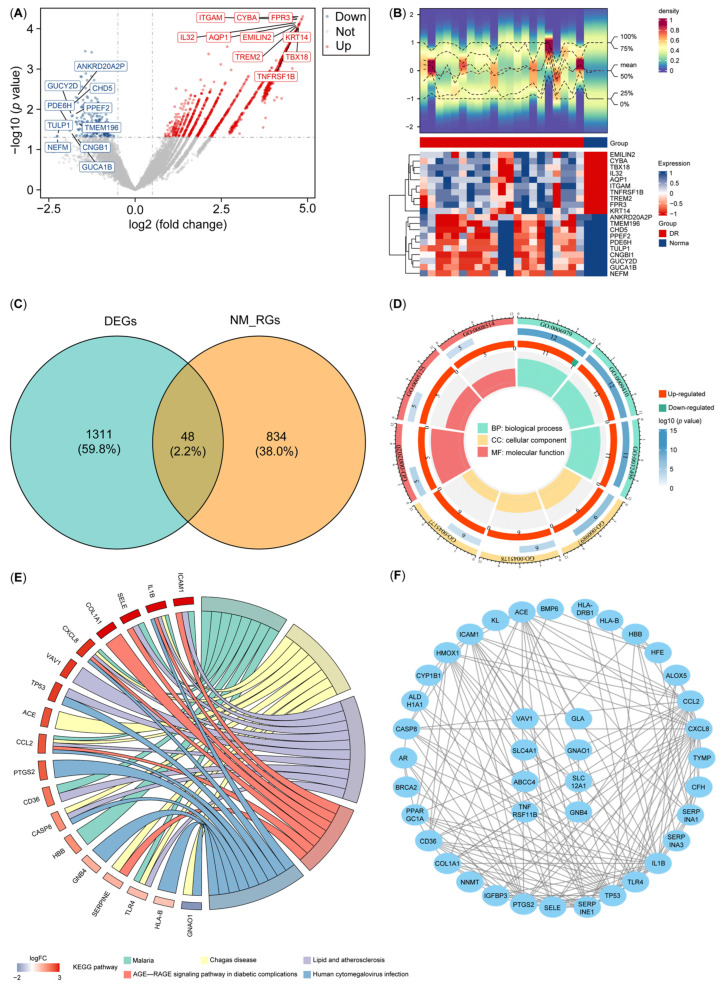
Functional enrichment analysis of DE-NMRGs. (**A**,**B**) Volcano plot and heatmap of DEGs. (**C**) Venn diagram of DEGs and NM-RGs. (**D**,**E**) Circle plot of the GO and KEGG enrichment analysis. (**F**) Protein and protein interaction network of DE-NMRGs. NM-RGs—nucleotide metabolism-related genes; DEGs—differentially expressed genes; DE-NMRGs—differentially expressed NM-RGs.

**Figure 2 biology-14-00409-f002:**
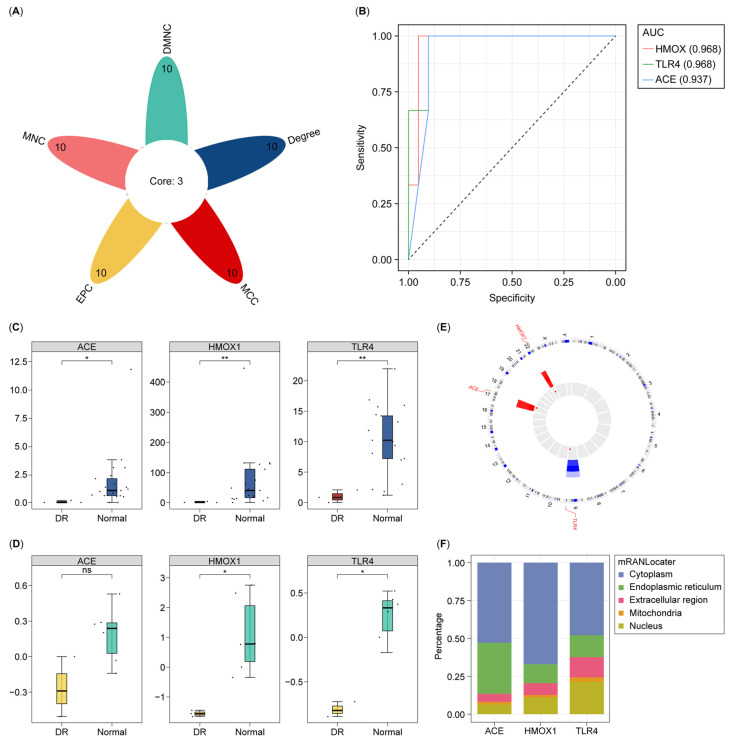
Identification and verification of biomarkers. (**A**) Venn diagram of the top ten genes in five algorithms. (**B**) ROC curves of three biomarkers. (**C**,**D**) The expression of three biomarkers within DR and controls in GSE102485 and GSE60436. (**E**,**F**) Chromosome and subcellular localization. ROC—Receiver Operating Characteristic. ns, *, **, indicate statistically significant differences at the *p* value > 0.05, *p* value < 0.05, *p* value < 0.01 levels, respectively.

**Figure 3 biology-14-00409-f003:**
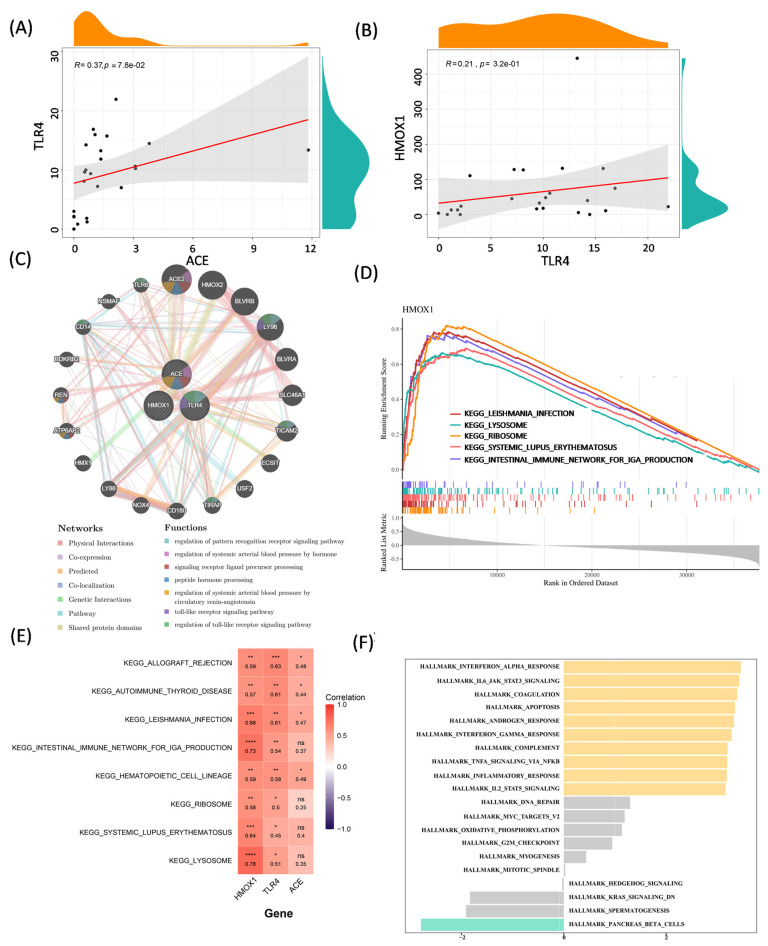
Correlation and functional analysis of biomarkers. (**A**) The correlation between ACE and TLR4. (**B**) The correlation between HOMX1 and TLR4. (**C**) The gene–gene interaction network. (**D**) The GSEA analysis of HOMX1. (**E**) KEGG enrichment analysis of three biomarkers. (**F**) GSVA. ns, *, **, *** and **** indicate statistically significant differences at the *p* value > 0.05, *p* value < 0.05, *p* value < 0.01, *p* value < 0.001 and *p* value < 0.0001 levels, respectively.

**Figure 4 biology-14-00409-f004:**
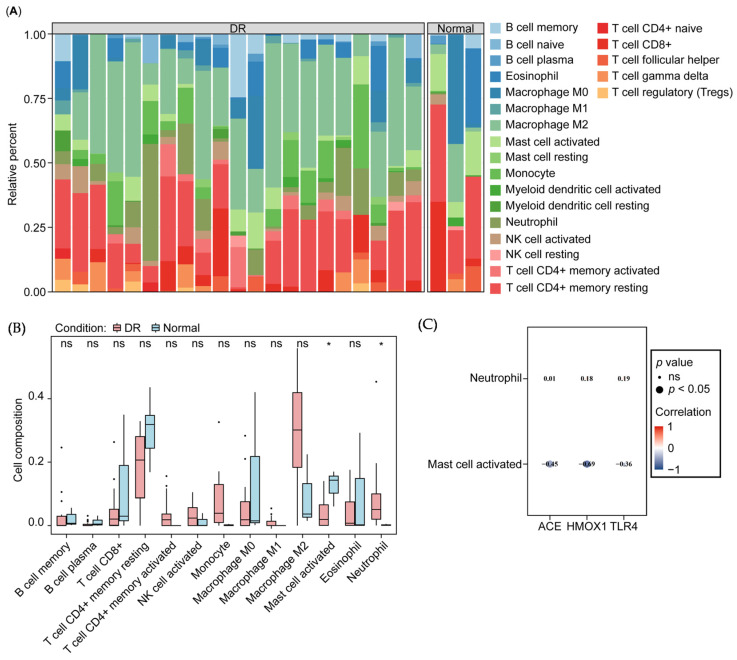
Immuno-infiltration analysis. (**A**) Stacked histogram of the percentages of immune cells drawn in GSE102485. (**B**) Variation in immune cell types between DR and controls. (**C**) The correlation between the three biomarkers with neutrophil or activated mast cell. ns, *, indicate statistically significant differences at the *p* value > 0.05, *p* value < 0.05 levels, respectively.

**Figure 5 biology-14-00409-f005:**
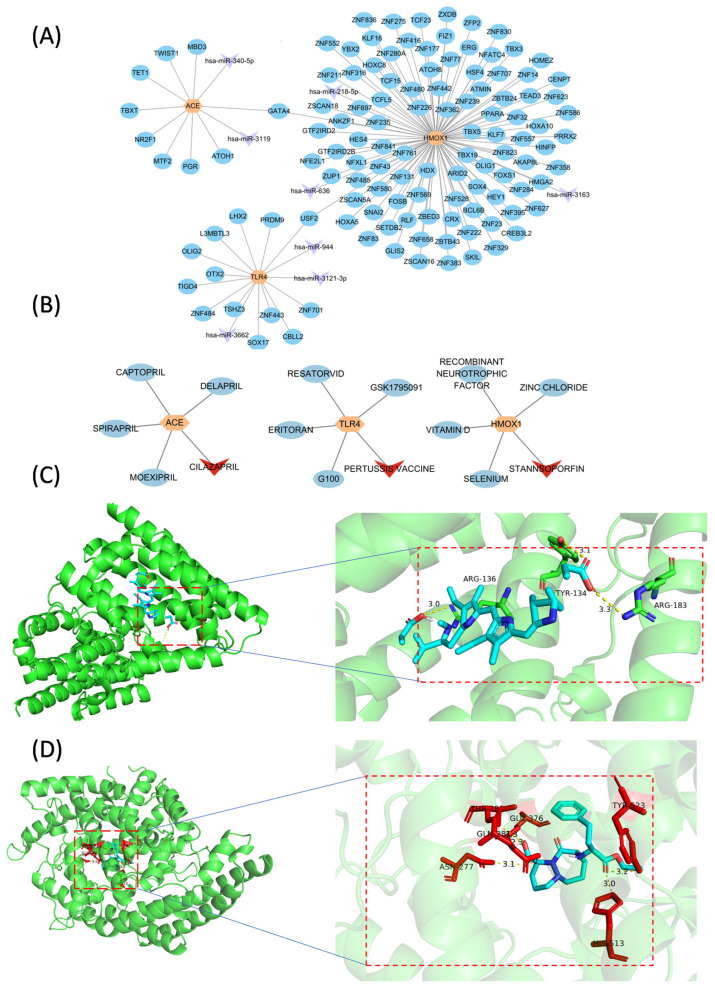
Molecular regulatory network and drug prediction. (**A**) TF-mRNA-miRNA network. (**B**) Drug–gene network. Blue dots represent the predicted drugs; red triangles represent the drugs with the highest prediction scores. (**C**) Molecular docking results of HMOX1–Stannsoporfin. (**D**) Molecular docking results of ACE–cilazapril.

**Figure 6 biology-14-00409-f006:**
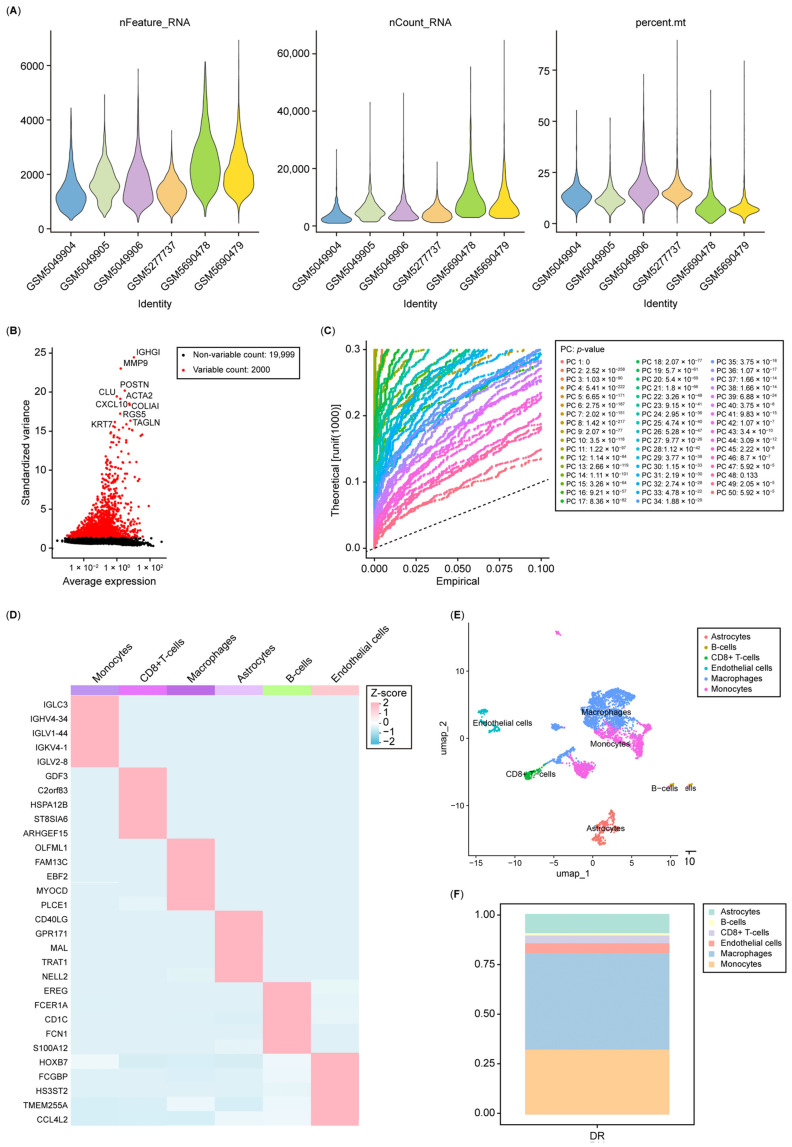
Analysis of single-cell RNA sequencing data of DR. (**A**) Distribution of gene counts per cell (nFeature_RNA), unique molecular identifiers (UMIs) per cell (nCount_RNA), and mitochondrial gene percentage (percent.mt) in the single-cell RNA-seq data. (**B**) Variance plot showing 21,999 genes across all cells, with the top 2000 highly variable genes highlighted in red. (**C**) Identification of the top 20 principal components based on a *p*-value threshold of <0.05. (**D**) Heatmap illustrating the expression of the top five marker genes across six detected cell clusters. (**E**) t-SNE clustering analysis grouped cells into six distinct types, with each color representing an annotated phenotype. (**F**) The proportion of each cell type.

**Figure 7 biology-14-00409-f007:**
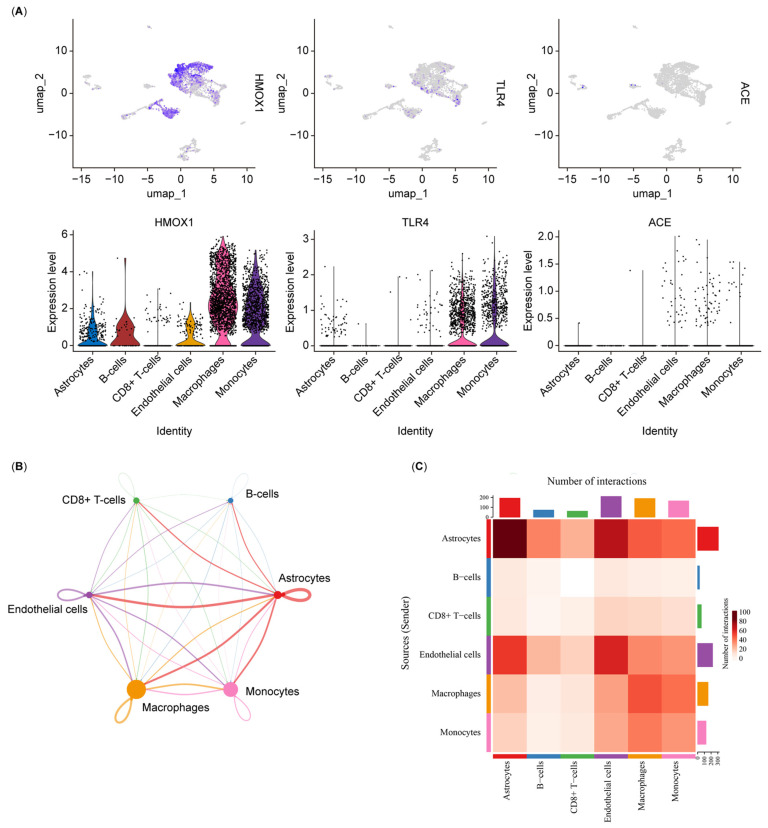
(**A**) Feature and violin plots showing the distribution of three biomarkers in various cell types. (**B**) Chord diagram illustrating the number of intercellular interactions based on ligand–receptor pairs. (**C**) Heatmap visualizing the intensity of ligand–receptor interactions between different cell types.

**Figure 8 biology-14-00409-f008:**
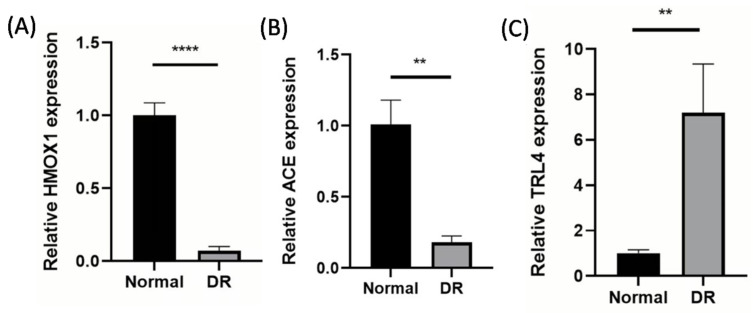
RT-PCR assays of HMOX1 (**A**), ACE (**B**), and TRL4 (**C**). ** and **** indicate statistically significant differences at the *p* value < 0.01 and *p* value < 0.0001 levels, respectively.

**Table 1 biology-14-00409-t001:** The primer sequences used in the study.

Primer	Gene	Sequence (5′ to 3′)
HMOX1 F	HMOX1	AAGACTGCGTTCCTGCTCAAC
HMOX1 R		AAAGCCCTACAGCAACTGTCG
TRL4 F	TRL4	AGACCTGTCCCTGAACCCTAT
TRL4 R		CGATGGACTTCTAAACCAGCCA
ACE F	ACE	CCACGTCCCGGAAATATGAAG
ACE R		AGTCCCCTGCATCTACATAGC

## Data Availability

All the RNA-sequencing data of diabetic retinopathy patients were acquired from the Gene Expression Omnibus database (GEO, https://www.ncbi.nlm.nih.gov/geo/query/acc.cgi, (accessed on 20 November 2024)).

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
