# Peer review of "Identification of Key Nucleotide Metabolism Genes in Diabetic Retinopathy Based on Bioinformatics Analysis and Experimental Verification"

_biology, 2025, doi:10.3390/biology14040409_

Round 1
Reviewer 1 Report
Comments and Suggestions for Authors
Dear Authors,
Manuscript summary: First of all, congratulations to the authors for their research on identifying three biomarkers, HMOX1, TLR4, and ACE, which are implicated in diabetic retinopathy (DR). The manuscript is well-written, with each section—Introduction, Materials and Methods, Results, and Discussion—thoroughly detailed. However, the authors should address the following comments:
- As part of related works, the authors must describe the currently known biomarkers associated with DR. Additionally, as part of this research, the authors should report the area under the curve (AUC) values for these established biomarkers and compare them with those of HMOX1, TLR4, and ACE. This comparison should be explicitly presented in the Results section and further elaborated upon in the Discussion section.
- It would significantly enhance the impact of this study if the authors could discuss any similar research that has previously demonstrated that one or more of these biomarkers (HMOX1, TLR4, and ACE) are associated with DR. If no prior studies or clinical evidence exist supporting these biomarkers' involvement in DR, this should be explicitly stated to highlight the novelty of the findings.
- Conclusion is too short. To improve the manuscript, I also recommend that, the authors rewrite the conclusion.
- Manuscript improvement comments
- Check line 87 formatting (the interferon - α response)
- Improve the legends and labels for most figures to enhance text readability. At 100% scale, readers should be able to clearly read all figure details. While Figures 4C and 8 are acceptable, the remaining figures require adjustments to improve labeling and legend size.
- Remove the half-page empty space between lines 315 and 316. Although this will likely be addressed during editing prior to publication, it would be beneficial for the authors to proactively correct it.
Reviewer 2 Report
Comments and Suggestions for Authors
The study investigated the differential expressed genes between the control and Diabetic Retinopathy group, specifically interested in genes that are involved in the nucleotide metabolism pathways. Critical genes involved in diabetic retinopathy were identified from the differential expressed genes using cytoscape and GSVA characterizations. Having established the core genes related to the diabetic retinopathy disease, next the group investigated the immune cells propotions in the diabetic patients by deconvoluting single cell populations using Cibersort program. Overall, this study constructs a transcription factor-mRNA-miRNA regulatory network for diabetic retinopathy and identifies potential biomarkers associated with the disease.
The findings of the study are interesting and the paper would be better presented in following issue can be addressed.
The Differential gene expression analysis was focused on the nucleotide metabolism related genes, woud it still helpful to perform the GO and KEGG analysis described in the figure 1E and 1F?
The figure 2D and 2E seemed redundant since the research was focused on differentially expressed genes at first, no doubt those gene expressions would be different.x
In line 288, the group deconvoluted the single cell populations from the bulk RNA-seq using cibersort program, the findings are quite interesting and would be further supported if the group can also validate with other programs, such as Music.
In line 317, the study proposed the potential mechanisms of the diabetic retinopathy. The protein protein interaction network does not seem appropriate here. It would improve the story if the author can use schematic pathway to explain the molecular mechanism.
In line 321, the abbreviation AA metabolism and OA was not explained at all.
Minor issue: GSE165784 was not described in the method section.
Comments on the Quality of English Language
The paper used the "DR" abbreviations so much, and the DR at the end the line are quite misleading to reader.
The Chinese-English writing style should be improved.
Reviewer 3 Report
Comments and Suggestions for Authors
The manuscript written by Wei Wang, Jianyang Gong “Identification of Key Nucleotide Metabolism Genes in Diabetic Retinopathy Based on Bioinformatics Analysis and Experimental Verification” presents valuable findings. In general, it is well written and needs some optimizations to be considered for publication in Biology.
Introduction:
The Introduction is well written with clear and structured ideas.
Line 86, please introduce AUC.
Line 87, GSVA without full spelling, then in line 94-95, In addition, gene set variation analysis (GSVA). Adjust that to have it in line 87 please.
For me, the introduction should ended with this sentence “This study aimed to identify and characterize key nucleotide metabolism-related 79 genes involved in DR through bioinformatics analysis and experimental validation.”
No need to describe the results here.
The paragraph from line 80 “Currently, research on the specific role..” to line 98 needs to be placed or divided between the beginning or the end of the discussion.
Mat & methods:
Fine. All information were correctly provided to the reader.
Please indicate the version of the GeneCards used in this paper. Same for STRING and the databases mentioned in 2.12.
Line 114 : In (log2FC| > 0.5, the first |is missing.
Results:
The results are robust, consistent and well presented.
There is an error in the listing of the elements in figure 3. gene-gene interaction network is C not B. Please adjust that.
Figure 4. You mentioned that (C) is “Correlation between biomarkers and immune cells”. But for me the graphic is showing the cell type variation between DR and controls. How do you link that to the biomarkers?
Line 354: “both ACE and HMOX1 were under-expressed significantly in DR group, while TLR4 was over-expressed significantly in DR group”, in this sentence we have ACE, HMOX1 ten TLR4, but in the figure below you changed the order. Please harmonize that to give the reader a clear idea. Please present down first, then up according to the text.
Discussion:
Line 361-362: reference is needed.
Line 364: Existing studies. Cite some of them, please
Line 489, “in vitro and in vivo experiments”. What kind of studies could you suggest in these cases ? Some examples ? Do you plan to carry out some of them in your lab or collaborate with other labs ?
Why you didn’t cite this paper 10.3389/fendo.2022.988506 as it used the same datasets as you and found TLR4 among the markers ? Same for https://doi.org/10.1371/journal.pone.0280548 for HMOX1 and TLR4.
Thank you.
Reviewer 4 Report
Comments and Suggestions for Authors
The article is beautifully written and presented. However, there is practically no practical significance in the work. The conclusion section needs to be expanded.
Round 2
Reviewer 2 Report
Comments and Suggestions for Authors
The GSE165784 has been descibed.
The title for 3.6 (line 325) has been fixed.
Author Response
Comments 1: The GSE165784 has been described. The title for 3.6 (line 325) has been fixed.
Response 1: Thank you for your positive feedback. We appreciate your suggestions.
Reviewer 3 Report
Comments and Suggestions for Authors
Dear authors, thank you for your answers.
I’m still have some comments.
Please, the legend of figure 3 is not correct. Check that and modify the text under the figure and in the list (line 636-637).
Figure 5, the (D) is missing.
The supplementary file of the revised version contains only the figures of the paper. I guess this was done by mistake. Please, you should verify the system when loading the revised version and keep the first supplementary file of the initial submission or resubmit it again with the revision #2. Only the final files will be considered for the publication, please ensure that all is good.
Thank you.
Author Response
Comment 1: Please, the legend of figure 3 is not correct. Check that and modify the text under the figure and in the list (line 636-637).
Response 1: Thank you for your valuable feedback. We have carefully reviewed and revised the legend of Figure 3 in the revised manuscript, ensuring that both the text under the figure and in the list (lines 636-637) are now accurate.
Comment 2: Figure 5, the (D) is missing.
Response 2: Thank you for pointing this out. We have added the missing (D) in Figure 5 in the revised manuscript..
Comment 3: The supplementary file of the revised version contains only the figures of the paper. I guess this was done by mistake. Please, you should verify the system when loading the revised version and keep the first supplementary file of the initial submission or resubmit it again with the revision #2. Only the final files will be considered for the publication, please ensure that all is good.
Response 3: Thank you for your comment. We will carefully verify the system and resubmit the correct supplementary files, ensuring that the initial supplementary file is included with the revised version for publication. We appreciate your guidance in ensuring that all files are complete.